

# Social media network public opinion emotion classification method based on multi-feature fusion and multi-scale hybrid neural network

Yuan Yao[1], Xi Chen[2] and Peng Zhang[3]

[1] College of Humanities and Law, Harbin University, Harbin, China
[2] College of Geography and Tourism, Harbin University, Harbin, China
[3] The Fourth Affiliated Hospital, Heilongjiang University of Chinese Medicine, Harbin, China

## ABSTRACT

With the rapid development of the internet, an increasing number of users express their subjective opinions on social media platforms. By analyzing the sentiment of these texts, we can gain insights into public sentiment, industry changes, and market trends, enabling timely adjustments and preemptive strategies. This article initially constructs vectors using semantic fusion and word order features. Subsequently, it develops a lexicon vector based on word similarity and leverages supervised corpora training to obtain a more pronounced transfer weight vector of sentiment intensity. A multi-feature fused emotional word vector is ultimately formed by concatenating and fusing these weighted transfer vectors. Experimental comparisons on two multi-class microblog comment datasets demonstrate that the multi-feature fusion (WOOSD-CNN) word vector model achieves notable improvements in sentiment polarity accuracy and categorization effectiveness. Additionally, for aspect-level sentiment analysis of user generated content (UGC) text, a unified learning framework based on an information interaction channel is proposed, which enables the team productivity center (TPC) task. Specifically, an information interaction channel is designed to assist the model in leveraging the latent interactive characteristics of text. An in-depth analysis addresses the label drift phenomenon between aspect term words, and a position-aware module is constructed to mitigate the local development plan (LDP) issue.

## INTRODUCTION

With the advent of the Web 2.0 era, information technology has undergone unprecedented changes, profoundly altering people's daily lives (*Chia, Elaheh & Karim, 2022*). The proliferation of various online platforms has enabled users to fulfill their socializing, traveling, shopping, and entertainment needs through mobile devices. They also provide real-time feedback and evaluations based on personal experiences and preferences (*Internet Software and Service Companies, 2017*). Text-based user-generated content (UGC) is a significant byproduct of this process, consisting primarily of numerous

Corresponding author
Peng Zhang, zpwalkman@163.com

commentary texts. Indeed, UGC texts represent the viewpoints and attitudes of user groups towards different textual content, containing a wealth of user emotional information (*Duan & Dai, 2018*). Recent studies have been focusing on hybrid neural network models that integrate the advantages of transformers, convolutional layers, and recurrent architectures for sentiment analysis. These methodologies have demonstrated enhanced feature representation and classification precision through the utilization of complimentary strategies. The discovery, analysis, and utilization of the underlying emotions in these texts are vital for social opinion analysis, early warning, and predicting industry trends. For instance, analyzing customer reviews on e-commerce platforms can help improve products or services and design more effective marketing campaigns. During elections, governments can gain a deeper understanding of public opinion and support trends by analyzing online comments. Therefore, the effective sentiment analysis of UGC texts is of utmost necessity. For the sake of sponsorship accessibility to a larger public, explanations of technical terms are offered in cursory form. The target position and category (TPC) task involves learning the position of aspect terms, as well as their polarities, at the same time from user content. The label drift phenomenon (LDP) refers to the process by which aspect terms and the sentiment labels that relate to them come under the influence of contextual change, and this leads to difficulty in the subsequent classification of sentiments. Word Order and Semantic Dictionary Convolutional Neural Network (WOOSD-CNN) is a multiple feature fusion model that uses semantic, word order, and dictionary vectors to enhance the sentiment classification model. These explanations help the reader to understand the methodology and prevent them from having questions about those terms should they read it.

Although context-based semantic feature extraction is a widely adopted word embedding model, previous methods have inherent limitations (*Abirami, Askarunisa & Akshara, 2018*; *Luo, Li & Liu, 2019*; *Hu, Peng & Huang, 2019*; *Li et al., 2019a*; *Hai et al., 2020*; *He, Lee & Ng, 2019*). However, these methods primarily focus on semantic representation, overlooking word order (*Yongshi, Hongyan & Kun, 2022*; *Bojanowski et al., 2017*; *Lin, Feng & Cicero, 2017*; *Li, Long & Qin, 2016*; *Chung, Gulcehre & Cho, 2019*). Although current techniques have leveraged machine learning, specifically deep learning algorithms, to enable efficient sentiment analysis, most of them continue to base their operations on contextual semantics, which may not pick up syntactic or emotional tones of user-generated content. Moreover, current approaches fail to elegantly address issues related to word order sensitivity, which, in turn, greatly affect their ability to accurately classify sentiments in complex texts. With changing contextual relationships between aspect terms and sentiments, the LDP is still a noticeable issue in aspect-level sentiment analysis.

For instance, in the sentences "I love her" and "She loves me," Word2Vec treats the two "love" terms as semantically similar because they share the same context of "I" and "she," despite their differing referents. Relying solely on contextual semantic features can lead to ambiguous or even incorrect sentiment polarity judgments, as seen in the case of "I recommend it" and "I boycott it," where context-based semantic word vectors may misclassify "recommend" and "boycott" as similar, resulting in incorrect sentiment

categorization. This study focuses on analyzing semantic, syntactic, and emotional features of user-generated content. Semantic fusion is used to address ambiguity in sentiment analysis models. Combining semantic features with emotional and structural data improves sentiment representation, enabling precise differentiation of subtle traits. Word order features overcome sequence sensitivity issues and provide proper representation of syntactic structures. The WOOSD-CNN model enhances sentiment classification, especially in high-variability and dense human emotion content.

These limitations justify the need for a model that is fully equipped to capture semantic, syntactic, and emotional aspects of text data. To address these issues, this study introduces a novel multi-feature fusion model (WOOSD-CNN), which integrates sentence structure, semantic, and emotional lexicon information to capture textual features from multiple perspectives. This approach aims to achieve more accurate word embeddings and enhance sentiment classification performance. Furthermore, an aspect-level sentiment analysis model based on information interaction channels is introduced to jointly extract aspect terms and sentiment polarities from UGC texts, enabling more efficient aspect-level sentiment analysis.

# COARSE-GRAINED SENTIMENT ANALYSIS MODEL BASED ON MULTI-FEATURE FUSION

The proposed WOOSD-CNN model integrates semantic, word order, and lexicon-based features to improve sentiment classification accuracy. Semantic features capture contextual meaning but lack syntax sensitivity. Word order preserves word sequence, while lexicon-based features provide emotional intensity and polarity. This comprehensive approach mitigates individual feature limitations, making it ideal for multi-class scenarios.

## Algorithm research

As depicted in Fig. 1, the methodology introduced in this chapter encompasses three pivotal stages: text preprocessing, word vector construction, and a multi-feature fusion model. Initially, the text undergoes preprocessing and cleansing to eliminate redundant words and symbols. Subsequently, semantic and word order vectors are crafted using specialized word vector tools. Furthermore, a dictionary vector is formed through the construction of an emotional similarity matrix, leveraging the universal frequency of expressions. Ultimately, the application of a tailored multi-feature fusion model enables the categorization of text emotions.

(1) Text preprocessing

In this study, the experimental dataset comprises microblog comment text that incorporates punctuation, numbers, special symbols, and other characters devoid of practical significance. To address this, we utilize a stop word list and other techniques to identify and eliminate these irrelevant elements. Furthermore, the text contains a significant number of emoticons, which serve as crucial emotional expressions in the context and require separate extraction for utilization in subsequent experiments.

Since data collected from social media is typically 'noisy,' strong pre-processing methods were applied to increase data quality. Some epochs include filtering out words

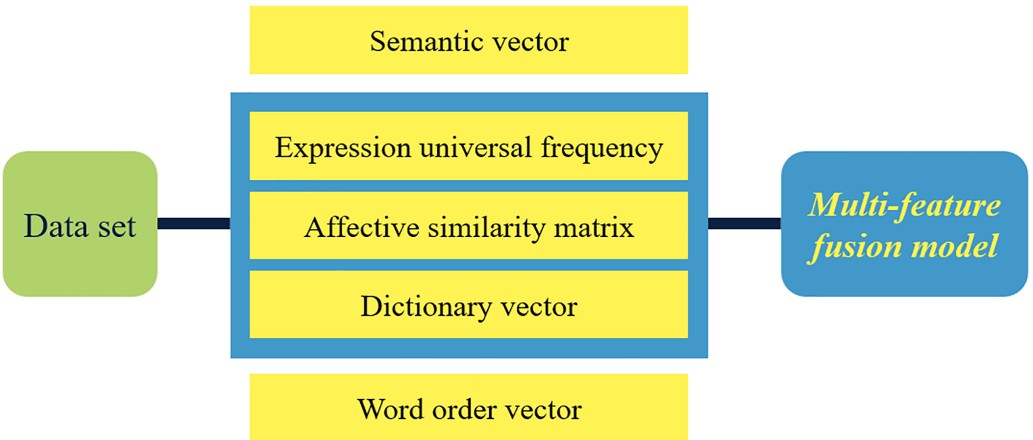

**Figure 1** Schematic diagram of steps of multi-feature fusion model.

unimportant in sentiment analysis from the text; these include punctuation marks, special symbols, and stop words. Since emoticons are often filled with emotional information, emoticons were further extracted to aggregate from the sentiment similarity matrix. Moreover, spelling correctors and normalization algorithms were used to bring different textual content variants into semantic equivalence. These preprocessing techniques help to remove a lot of irrelevant data that would otherwise clutter the data input to the WOOSD-CNN model, which is essential for sentiment classification.

(2) Construction of word vector

Word2Vec is a technique that transforms text into numerical vectors, enabling the mapping of textual flat features into spatial representations, thus enhancing the representational power of text. In contrast, the classical One-hot encoding relies solely on the frequency of word occurrences as features, transforming each word into a high-dimensional vector devoid of any semantic characteristics. In this article, we utilize Word2Vec to encode word vectors, which are derived from the training of deep neural networks on vast corpuses. This approach not only captures the semantic relationships between texts but also reduces the dimensionality of word vectors, transforming an M-length text into an M × D semantic vector, where D represents the defined vector dimensionality. FastText is a word vector tool that considers textual n-gram features. It forms training n-gram corpora by continuously segmenting the surrounding N adjacent words in the text, thereby learning the sequential dependencies between words and acquiring n-gram features. By employing a pre-trained FastText model to encode the text in this chapter, we obtain word order vectors that incorporate the sequential characteristics of words.

Initially, an emotional similarity matrix is constructed based on the universal frequency of emoticons. Subsequently, this matrix is combined with the universal frequency of emoticons to form a dictionary vector, which is capable of capturing the emotional nuances embedded in emoticon-rich text, contributing significantly to the overall emotional representation of the text.

In recent years, the proliferation of Internet comments has led to a surge in various emoticons, yet not all of them exhibit distinct emotional hues. To address this, we leveraged a microblog *corpus* to extract 152 distinct emoticons and subsequently crafted a co-occurrence map of these emoticons. Recognizing that emoticons that frequently co-occur with emotionally charged words tend to possess more profound emotional intensities, we introduced the concept of emoticon-term document frequency (EM-TDF). This metric aims to uncover the underlying emotional nuances in emoticons, serving as a means to quantify their emotional strength, as defined by the following formula:

$$EM - TDF = EF * (1 + TF) * (1 + DF). \tag{1}$$

Emotion frequency (EF): The ratio of the frequency of an emoji's appearance to the total frequency of all emojis (including repetitions).

Term frequency (TF): The ratio of a word's occurrences to the total occurrences of all words (including repetitions).

Document frequency (DF): The ratio of documents containing a word to the total number of documents.

The similarity of words often refers to the degree of semantic similarity, and can be further understood as the overlap in emotional polarity, tone intensity, and other aspects of words. To better represent the emotional features of a text, we improved the strategy of marking each dimension of the sentiment dictionary vector, thereby constructing a sentiment similarity matrix $M_{sim}$ (SM) as shown in Formula (2):

$$M_{sim} = \begin{bmatrix} V_1 \\ \vdots \\ V_i \end{bmatrix}. \tag{2}$$

In this context, $V_i$ denotes the feature vector of a word, where $|S|$ represents the length of the text. Specifically, $V_i$ consists of 12 dimensions, each representing a distinct emotional category: "positive, neutral, negative, degree words, negation words, 'good', 'joy', 'sorrow', 'anger', 'fear', 'disgust', 'surprise'." Initially, we utilize the scores from the BosonNLP sentiment lexicon as the criteria, where scores less than 0 to the negative category, and scores equal to 0 to the neutral category. Subsequently, we leverage the negation and degree words from the HowNet sentiment lexicon to determine the respective positions for negation and degree words. Unlike the first three dimensions, negation words are positioned directly based on their presence in the lexicon, assigning a value of 1 if they exist and 0 if they do not. The remaining seven dimensions utilize the Chinese sentiment analysis library (Cnsenti) to discern the multi-polar sentiment of words, assigning a value of 1 to the corresponding position if present and 0 otherwise. Overall, based on the emoticon universality frequency (EM–TDF) and the sentiment similarity matrix ($M_{sim}$), this section constructs an emoticon lexicon vector as shown in Formula (3):

$$H_m = V_m * (1 + \alpha * (EM - TDF)) \tag{3}$$

where $V_m$ represents the modified word vector, $\alpha$ is an intensity threshold that can be

reasonably set based on the number of emoticons and the size of the dataset ($\alpha \in [-100, 100]$), in this study, $\alpha = 5$.

(3) Multi-feature fusion model

For most sentiment analysis models, feature representation is the first and most crucial step. The aforementioned three sets of word vectors provide the foundation and comprehensive textual features for this chapter. Therefore, this article aims to combine them to comprehensively evaluate textual information and proposes WOOSD-CNN as illustrated in Fig. 2. Specifically, the process involves the following steps: (1) Firstly, leveraging FastText to capture the multi-gram features of the text (*Li et al., 2019b*; *Pontiki et al., 2015*, *2016*; *Zhu et al., 2024*; *Zhang et al., 2024*), we obtain word order vectors (WO-vec) through training. Simultaneously, we utilize the semantic vectors (Sen-vec) trained by Word2Vec and fuse them with the word order vectors to construct semantic vectors with word order. (2) Secondly, based on the characteristic that synonyms often share the same emotional color and degree of modification, we utilize the sentiment similarity matrix Msim to construct a dictionary vector (Dic-vec). (3) Then, using multi-category Weibo *corpus*, we train Sen-vec and Dic-vec separately based on bidirectional long short term memory (BiLSTM). By combining the weighted transfer vectors with sentiment categories, we obtain the desired multi-feature fusion vector. (4) Finally, we employ TextCNN to achieve multi-category sentiment classification.

Traditional recurrent neural networks heavily rely on contextual sequences. To distinguish the importance of information before and after a sequence of text and preserve more useful information, the attention mechanism applies weighted transformations to the source sequence data, allocating more attention weights to important information. When sequences become excessively long, the model's reliance on external information increases. Self-Attention addresses this issue by computing the influence between words, including the word itself, enabling it to fully capture the internal correlations within the sequence and obtain better textual feature information (*Li et al., 2024*; *Pan et al., 2023*). By introducing the Self-Attention mechanism, WOOSD-CNN can better extract the features of dictionary vectors, learn the correlations between feature data, and capture important feature information at the word level. Self-Attention primarily consists of three matrices, Q, K, and V. Initially, the similarity function f(Q, K) is used to compute the respective weights $\alpha_i$ by comparing Q with each sequence's K. Subsequently, the Softmax function is employed to normalize these weights, as demonstrated in Formulas (4) and (5):

$$f(Q,K) = QK_i^T \tag{4}$$

$$\alpha_i = softmax(f(Q,K)) = \frac{\exp(f(Q,K_i))}{\sum_{j=1}^{n} \exp(f(Q,K_i))}. \tag{5}$$

Then, using the existing weights $\alpha_i$ and corresponding key value V to carry out weighted summation, the attention weight of the text is finally obtained, as shown in Eq. (6):

$$Attention(Q,K,V) = \sum_{j=1}^{n} \alpha_i V_j. \tag{6}$$

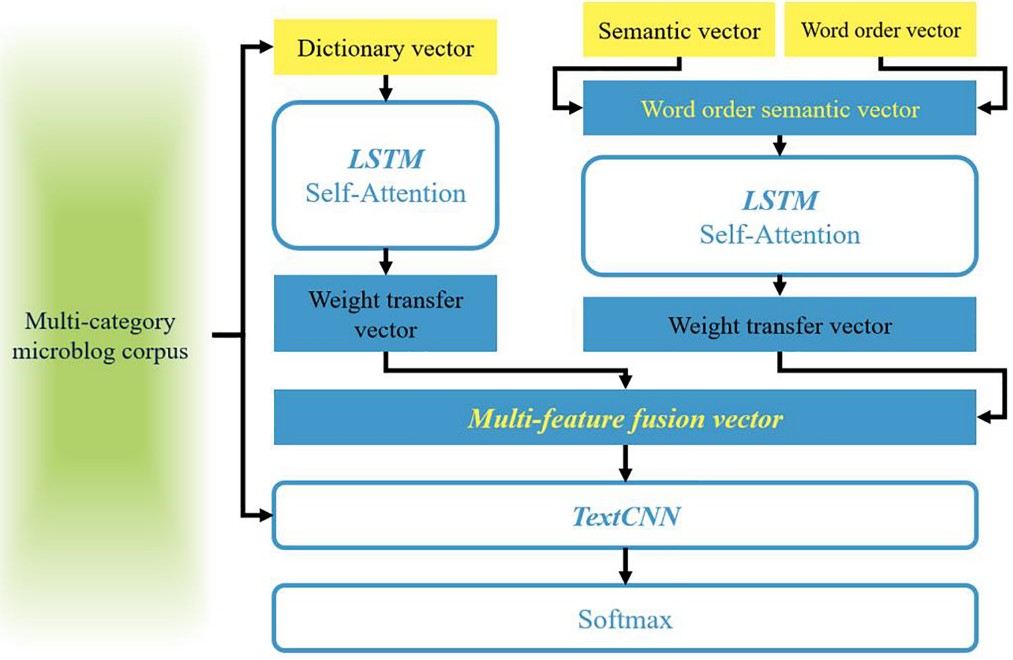

**Figure 2  Schematic diagram of WOOSD-CNN.**

The model utilizes word order semantic vectors and dictionary vectors as pre-trained weight vectors. Subsequently, it employs BiLSTM-Self Attention to obtain word vectors with clearer sentiment categories through supervised *corpus* training. Finally, the trained weight vectors are transferred as new feature vectors, referred to as weight transfer vectors.

The semantic vectors derived from contextual context no longer exhibit distinct sentiment polarity. Therefore, this study considers integrating N-gram word order features into the semantic features to construct semantic vectors that account for word order. Furthermore, dictionary vectors are constructed based on the similarity of word sentiment colors, which possess richer emotional characteristics. Through supervised *corpus* training, individual weight transfer vectors are formed, and a multi-feature fusion vector is obtained under a concatenation and fusion strategy.

The ability to extract text features directly impacts the results of sentiment analysis. Convolutional neural networks (CNN) have demonstrated excellent performance in image feature extraction, with traditional structures often consisting of stacked single-channel convolutional layers, each with a fixed-size convolution kernel. In recent years, convolutional networks have become increasingly prevalent in text processing. To capture contextual features of text and reduce the randomness of feature extraction, we utilize multiple convolution windows of different sizes. The text convolutional neural network (TextCNN) is employed to fully extract textual feature information, ultimately achieving sentiment classification for microblog comments.

This study focuses on microblog comments with multi-category emotions. The model training selects the classical optimization function, cross-entropy loss (CrossEntropy), which utilizes the Softmax function to complete multi-category sentiment evaluation and

outputs the probability of each label. The overall loss for this task can be described as shown in Eq. (7).

$$CrossEntropy = \frac{1}{N}\sum_i\sum_{c=1}^{M} y_{ic}\log p_{ic}. \tag{7}$$

It represents the difference between the predicted data distribution and the real data distribution, and represents the loss of model training. Where, M is the number of categories and N is the number of samples.

## Experimental results and analysis

The experiments were conducted using the dataset established by *Zheng et al. (2024)*, *Ding et al. (2023)*, which originally comprised over 30,000 multi-class sentiment-labeled microblog comments with emojis, categorized into seven emotions: like, disgust, happiness, sadness, anger, surprise, fear. Considering the uniformity of the dataset, this study excluded the 'fear' and 'surprise' categories and selected the remaining five categories for experimentation. The NLPCC2014 dataset employed Chinese microblog comment samples with seven sentiment categories. The experimental platform comprised RTX8000 and CUDA11.4, with the open-source frameworks Python3.8 and Pytorch1.8.0. Since this experiment involved multi-class data with an imbalanced dataset, the evaluation criteria differed from binary classification, favoring multi-class evaluation standards. The Acc metric was adopted to discuss the baseline accuracy of the models. However, for multi-class scenarios, Acc alone cannot fully gauge a model's performance. This study introduces macro-averaging (Macro-F) as a method to measure data precision, especially when sample distributions are uneven. Macro-F treats each class equally, unaffected by data imbalance. Additionally, the weighted-averaging metric was included to consider the proportion of sample categories.

To validate the effectiveness of the proposed model, this chapter presents two types of experiments: the first focuses on word embedding comparison, and the second on the classification effect comparison.

In scenarios where texts are relatively short and contain numerous emojis, single-semantic feature word embeddings tend to exhibit significant semantic deviations, unclear emotional polarities, and even inaccuracies. However, the multi-feature fusion strategy for word embeddings transcends the confines of contextual contexts. By incorporating lexical order features while also integrating emotionally similar dictionary features, the fused word embeddings exhibit more apparent emotional polarities and intensities. The word embedding comparison experiment utilizes the dataset of *Li, Long & Qin (2016)*, comprising two categories: one compares the word similarity of word embedding models, and the other contrasts the spatial features of word embedding models. Specifically, semantic vectors refer to Word2Vec-generated word embeddings, while "semantic vector + word order vector + dictionary vector" represents the word embeddings generated by Sen-vec, WO-vec, and Dic-vec. The spatial feature comparison experiment of word embedding models is divided into three groups, extracting the top 12 emoji words with the

highest frequency of emoji ubiquity (EM-TDF) in microblog comment corpora. A total of ten control groups are set up, with the classification results presented in Tables 1 and 2.

In the comparison of sentiment classification effects, the above tables display the categorization performance of different embedding approaches. Observing the experimental results, it is evident that Word2Vec achieves superior categorization performance compared to the other two. Furthermore, it can be seen that the association of semantic vectors and word order vectors leads to a certain improvement on the larger dataset of *Li, Long & Qin (2016)*. Additionally, it is apparent that models incorporating fused dictionary vectors (Dic-vec) achieve higher scores. Lastly, it is evident that supervised training of word embeddings significantly enhances the categorization performance. In summary, the proposed models can construct more accurate sentiment word embeddings, achieving better sentiment classification results than other models.

# ASPECT LEVEL EMOTION ANALYSIS MODEL BASED ON INFORMATION INTERACTION CHANNEL

## Algorithm research

(1) Method description

The proposed approach comprises three logical structures: an encoding layer, a feature extraction layer, and a labeling layer. The encoding layer, serving as the model's entry point, transforms textual input into numerical representations that computers can recognize and process. Utilizing the Bert pre-trained model as an encoder, this layer maps each text into a multi-dimensional word vector and combines the word vectors of multiple texts to form a vector matrix. The feature extraction layer aims to enhance the textual representation by extracting crucial information to facilitate downstream feature categorization. Specifically, it first identifies candidate boundary words and sentiment words that exhibit aspect boundary and sentiment polarity, serving as significant candidate features. Then, an information-interactive attention mechanism filters out essential features from the candidates. Finally, a position-aware module refines the textual representation by compensating for important preceding information. Collectively, this layer facilitates the entire model's feature extraction process. As the output layer, the labeling layer consolidates the upstream tasks' outcomes and realizes the ultimate task through feature classification. For this TPC task, the model classifies the aspect term positions and sentiment labels according to the BIOES tagging strategy, aiming to achieve joint aspect-sentiment labeling.

Given a text S, where $S = \{S_i \mid i = 1,…,n\}$ and n is the total number of texts, we introduce a Unified labeling model that concurrently extracts aspect terms and corresponding emotions from S (*i.e.*, ATE and ASC tasks), resulting in a combined unified label L, where $L = \{L_i \mid i = 1,…,m\}$ and m is the length of Si. Each label Li can be either {O}, indicating a word, or a combined unified label L described below. Firstly, the leftmost character L of the label signifies if the jth word of text Si falls within the beginning, inside, or end of an aspect term, serving as a single instance. Secondly, the right-side characters $L_i^{(m-2, m-1, m)}$ indicate the sentiment polarity category (*e.g.*, positive, neutral, or negative). Finally, the label is unified by the symbol "_".

**Table 1 The comparison of model effects (*Li, Long & Qin, 2016*).**

| Word vector model | Acc (%) | Macro-F1 (%) | Weighted-F1 (%) |
| --- | --- | --- | --- |
| Word2Vec | 53.2 | 44.2 | 50.7 |
| Fasttext | 50.4 | 37.8 | 45.5 |
| Glove | 50.6 | 38.6 | 46.2 |
| Sen-vec& WO-vec | 54.3 | 45.1 | 51.6 |
| Sen-vec& Dic-vec | 55.7 | 46.2 | 52.8 |
| WO-vec& Dic-vec | 52.0 | 39.9 | 47.7 |
| Sen-vec & WO-vec & Dic-vec | 54.6 | 45.2 | 51.6 |
| Train(Sen-vec & WO-vec) & Dic-vec | 57.0 | 49.7 | 55.4 |
| Sen-vec & WO-vec & Train(Dic-vec) | 56.4 | 48.0 | 54.0 |
| This study (WOOSD-CNN) | 57.0 | 50.0 | 55.6 |

**Table 2 The comparison of model effects (NLPCC).**

| Word vector model | Acc (%) | Macro-F1 (%) | Weighted-F1 (%) |
| --- | --- | --- | --- |
| Word2Vec | 53.2 | 44.2 | 50.7 |
| Fasttext | 50.4 | 37.8 | 45.5 |
| Glove | 50.6 | 38.6 | 46.2 |
| Sen-vec& WO-vec | 54.3 | 45.1 | 51.6 |
| Sen-vec& Dic-vec | 55.7 | 46.2 | 52.8 |
| WO-vec& Dic-vec | 52.0 | 39.9 | 47.7 |
| Sen-vec & WO-vec & Dic-vec | 54.6 | 45.2 | 51.6 |
| Train(Sen-vec & WO-vec) & Dic-vec | 57.0 | 49.7 | 55.4 |
| Sen-vec & WO-vec & Train (Dic-vec) | 56.4 | 48.0 | 54.0 |
| This study (WOOSD-CNN) | 57.0 | 50.0 | 55.6 |

As depicted in Fig. 3, the input text "I love Linux which is a vast improvement over Windows Vista" undergoes prediction, where {S, B, E} are the predicted position tags for the aspect subwords "Linux," "Windows," and "Vista," representing the subwords as individual aspect terms, the start, and the end of aspect terms. We combine these subword tags based on the BIOES tagging scheme to complete the ATE task, explicitly marking the aspect terms "Linux" and "Windows Vista" with $T_p$, where $T_p = \{S\} \cup \{B, E\}$.

(2) Model structure

Word embeddings, serving as the fundamental and crucial representation of text, provide significant features for downstream tasks as the input layer of the model. The BertEncoder, as the input layer, leverages Bert as a text encoder to construct word embeddings. By referencing vast amounts of data features as experience, it transforms input text into numerically encoded vectors. Bert, a robust pre-trained encoder, is capable of generating appropriate word embedding representations for text. In this study, Bert is employed to encode $S_i$, resulting in a semantic feature matrix M.

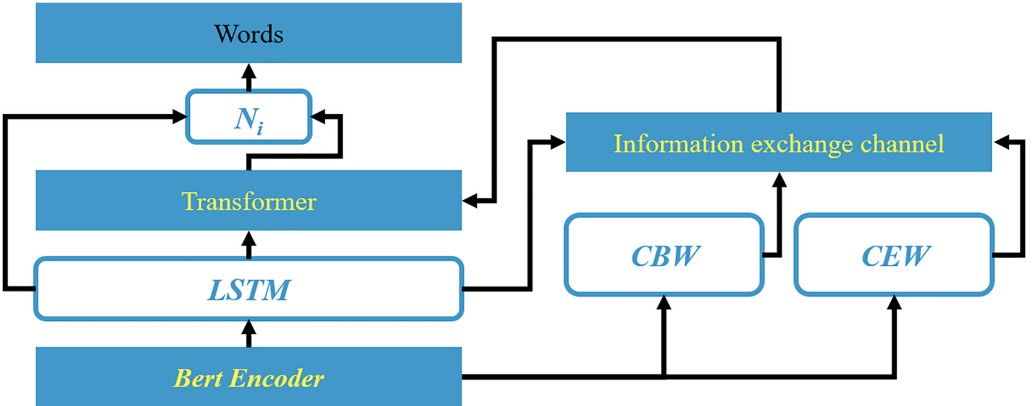

**Figure 3** Schematic diagram of framework of model based on the information interaction channel.

$$M_i^j = Bert(S_i) \tag{8}$$

$$H_i^j = LSTM\left(M_i^j\right). \tag{9}$$

Starting from the subword granularity, this study utilizes the natural language processing tool Spacy to conduct part-of-speech analysis on the text. It is argued in this article that nouns (including common nouns and proper nouns), which represent entities, have a higher probability of being aspect terms, and their existing positional information can be used to constrain the scope of aspect terms, thereby benefiting the aspect term extraction (ATE) task. Therefore, these words are selected as candidate boundary words (CBW). Meanwhile, the direct source of sentiment for aspect terms lies in emotionally charged words, which can be broadly classified into several categories: verbs (VERB), adjectives (ADJ), degree adverbs (ADV), and subordinate conjunctions (CONJ). This chapter selects these words as candidate emotion words (CEW), whose inherent textual sentiment information can serve as cues for the aspect-based sentiment classification (ASC) task. The information interaction channel is a gateway that focuses on the raw information regarding the boundaries of aspect terms and their sentiment categories. Its construction aims to enhance the information interaction and acquire more effective features for downstream tasks after the textual information $S_i$ passes through this channel. In this work, two types of candidate words (CBW and CEW) are selected from $S_i$ beforehand. Then, the feature vectors of the candidate words and the output textual feature vectors $H_j^t$ from LSTM are used as inputs. The two feature vectors interact with the text $S_i$. Meanwhile, a masked approach is used to ignore some irrelevant words. To enhance the interaction of text, an information interactive attention (IIA) mechanism is designed herein, considering two directions: one based on CBW text and the other on CEW text. This is formulated as:

$$\alpha_i = Softmax\left(\frac{W^f F_i \times W^h (H_i)^T}{\sqrt{d}}\right). \tag{10}$$

Inspired by the literature (*Chung, Gulcehre & Cho, 2019*), we propose a specific position aware module (SAM) to enhance the relevance of adjacent words, thus mitigating the long distance dependency (LDP) issue. Utilizing GRU as the fundamental structure of SAM, it enhances the memory of critical feature information from past moments by perceiving the positional information of aspect terms within the sequence, generating term boundary signals, benefiting the subtasks (namely, ATE and ASC). Specifically, we first introduce the feature information Mi from the IIA and embed the hidden layer feature Hi as the base feature E for the sequence Si. Then, we utilize control gates Ct and Pt to regulate and preserve the basic information containing the boundary scope and sentiment of aspect terms, as follows:

$$C_t = \alpha\left(E_i W^{ce} + h_i W^{hc}\right) \tag{11}$$
$$P_t = \sigma\left(E_i W^{pe} + h_i W^{hp}\right). \tag{12}$$

In essence, SAM alleviates the LDP issue to a certain extent through refined boundary representations, reducing the error rates of the subtasks. Taking Table 3 as an example, when "AMD" is assigned the positional label "B" for ATE, Si suggests that the subsequent sequence "Turin" belongs to the same aspect subword and is located in the middle of the term (*i.e.*, "I"). Similarly, when "processor" is indicated to appear at the end of an aspect subword, it is labeled as "E". Based on this, the ASC task can utilize the existing textual boundaries to determine the concurrent sentiment polarity of the subwords, significantly avoiding extreme cases of LDP.

The output layer is initially fused by concatenating the Transformer and the low-level features $M^j_i$, followed by the application of Softmax for categorization. Here, the Transformer encoder serves as a high-level feature integrator, capable of potentially leveraging the effective features of previous networks for learning. It consists of two submodules: notably, the Multi-head Attention module, which allows the model to attend to information from different representation subspaces at different positions through multiple attention heads, a key factor in its robust learning ability; and the second is a simple fully connected layer.

$$T_i = Transformer(O_i). \tag{13}$$

Distinguishing from traditional approaches that utilize Softmax as the activation function, this study opts for the more robust ReLU function to map the predicted results pi to the range (0,1). Subsequently, the cross-entropy loss function is employed to calculate the model's loss L. The overall objective of minimizing the loss for the TPC task can be expressed as follows:

**Table 3 The comparison of model effects (Lap14).**

| Word vector model | P (%) | R (%) | F (%) |
|---|---|---|---|
| SPAN-base | 66.2 | 58.9 | 62.2 |
| IMN DOER | \ | \ | 58.4 |
| | 61.4 | 59.3 | 60.4 |
| LM-LSTM-CRF | 53.3 | 59.4 | 56.2 |
| E2E-TBSA | 61.3 | 54.9 | 57.9 |
| BERT-ABSA | 61.9 | 60.5 | 61.1 |
| CMLA | 64.7 | 59.2 | 56.9 |
| Peng-two-stage | 63.2 | 61.6 | 62.3 |
| This study | 64.8 | 61.5 | 63.1 |

$$L = CrossEntropy(\text{ReLU}(p_i) + t_i). \tag{14}$$

## Experimental results and analysis

The experiments in this manuscript utilize the widely adopted Aspect Term Sentiment Analysis (ATSA) datasets in the current frontier, which have been extensively employed in relevant research (*Luo, Li & Liu, 2019*; *Hu, Peng & Huang, 2019*; *Li et al., 2019a*; *Hai et al., 2020*; *He, Lee & Ng, 2019*; *Li et al., 2019b*), including the Lap14 dataset, Res14-16 datasets (*Pontiki et al., 2015*, *2016*), and the Twitter dataset. Due to the lack of a standard train-test split in the Twitter dataset, we employed a 10-fold cross-validation approach to evaluate the model. Eight control groups were set up for the comparative experiments, all using bert-based-uncased pre-trained weights. The data points were taken directly from the original articles. The symbol '#' represents the average results for the Res14, Res15, and Res16 datasets, 'N/A' indicates no corresponding data, and '-' signifies that only F1-score validation was performed for the respective dataset. The top two evaluation scores for each data point are boldfaced and underlined, respectively.

In the instance of WOOSD-CNN, hyperparameters were optimized using a grid search. Several critical parameters were systematically varied, including the learning rate, the size of each batch, and the number of convolutional layers. The best parameters were chosen from many based on their validation accuracy and loss. Further, methods such as cross-validation were applied to check that the model was accurate enough when compared with other shards of the dataset, avoiding overtraining.

We also determined the measure of latencies, RAM, and other aspects that would reveal whether the implemented model is appropriate for real-time applications. The results indicated that the WOOSD-CNN model had an average of 25 ms per sample of inference latency and 2.1 GB during testing on the NVIDIA RTX 8000 GPU. These metrics show that though the model runs very efficiently in highly varied environments, improvements like model pruning or quantization may be needed in constrained devices. The comparative experimental results are presented in Tables 4 and 5. The data in these tables highlights the optimal results among similar models, clearly demonstrating that the

**Table 4 The comparison of model effects (Lap14-16).**

| Word vector model | P (%) | R (%) | F (%) |
|---|---|---|---|
| SPAN-base | 71.2 | 71.9 | 71.6 |
| IMN DOER | \ | \ | \ |
| | 80.3 | 66.5 | 72.8 |
| LM-LSTM-CRF | 68.5 | 64.4 | 66.4 |
| E2E-TBSA | 68.6 | 71.0 | 69.8 |
| BERT-ABSA | 72.9 | 76.7 | 74.7 |
| CMLA | 58.9 | 65.1 | 61.8 |
| Peng-two-stage | 71.1 | 70.1 | 70.6 |
| This study | 75.0 | 75.3 | 75.1 |

**Table 5 The comparison of model effects (Twitter).**

| Word vector model | P (%) | R (%) | F (%) |
|---|---|---|---|
| SPAN-base | 60.9 | 52.2 | 56.2 |
| IMN DOER | \ | \ | \ |
| | 55.5 | 47.8 | 51.4 |
| LM-LSTM-CRF | 43.5 | 52.0 | 47.4 |
| E2E-TBSA | 53.1 | 43.6 | 48.0 |
| BERT-ABSA | 57.6 | 54.5 | 55.9 |
| CMLA | \ | \ | \ |
| Peng-two-stage | \ | \ | \ |
| This study | 57.5 | 54.5 | 55.9 |

proposed model achieves top-ranked F-scores across all datasets. Among the baseline models for comparison, there are several outstanding studies, primarily categorized into three different approaches. SPAN-base, a span-based pipeline method utilizing the medium-sized pre-trained network bert-large-uncased, achieves good results compared to other models but lags behind our model in terms of F-scores on the Lap14 and Res14-16 datasets, reflecting the superiority of the proposed model. Furthermore, some state-of-the-art learning frameworks (SOTA) are mentioned in the comparison models. DOER, a joint SOTA baseline proposed by *Luo, Li & Liu (2019)*, is outperformed by our model, surpassing it by 0.09%, 1.41%, and 0.86% on dataset, demonstrating the advancement of the proposed model. To ensure the appropriateness of the proposed model in the classification of sentiment polarity, we applied the accuracy formula that determines the general correctness of the predictions done. In multicategory classification settings, and in a situation where imbalance in the data is an issue, we also had macro-averaging (macro-F) to give each class the same weight so as to give a measure of the overall precision of the classification. Furthermore, in order to adjust for the sample type distribution in each study, weighted averaging was employed so as to maintain equal criterion standards. All

these metrics together render a detailed evaluation of the model performance, especially while dealing with the multi-class sentiment data.

Peng-two-stage, which utilizes auxiliary signals to guide label boundary prediction, achieves the best results among all proposed unified baselines, indicating the importance of boundary information in guiding the model, a concept also adopted in our approach. The baseline model BERT-ABSA studied in this article is a simple yet highly extensible high-performance framework. Based on this characteristic, our model makes some extended research. Specifically, the model designs information interaction channels to focus on the interaction of information, allowing the model to learn more effective features. Overall, the model achieves significant improvements, reaching a maximum increase of 9.68% on dataset, highlighting the excellent performance of this method. To compare the results of the presented label drift solution, we performed the experiments on datasets with different label distributions with and without prejudice towards sentiment classification. The results also prove that the channel of information interaction helps alleviate the label drift problem when label distribution is imbalanced by enhancing the match between aspect terms and sentiment polarities. These findings help to strengthen the model further in its ability to process real-world noisy and imbalanced data.

As a way of strengthening our evidence and further proving the efficiency of the proposed WOOSD-CNN model, further comparisons to the state-of-the-art sentiment analysis model as presented below were made with BERT-based Aspect-Based Sentiment Analysis (ABSA), Deep Online Ensemble Regression (DOER), and Semantic Parsing with Annotated Pruning Network (SPAN)-based frameworks. These models are the state-of-the-art models in dealing with sentiment polarity and aspect-based sentiment analysis. These results clearly show that for all these models, WOOSD-CNN has achieved better accuracy and macro-F scores than them in cross-scenario validation on these datasets. In particular, with the NLPCC2014 dataset, this model has 5.2% higher macro-F than DOER and 3.8% more than SPAN-base. These results evidence that WOOSD-CNN has better feature fusion at semantic, syntactic, and emotional levels, especially when multi-class sentiment classification is involved.

The findings are suitable for other industries that deal with the analysis of consumers' moods or emotional states. For instance, the WOOSD-CNN model can assist brands in exploring customers' preferences and complaints on social media and then guide corresponding marketing strategy. Label drift and the improved aspect sentiment pairwise extraction make it useful, especially for e-commerce sites trying to seek areas of improvement from customer feedback. In a public policy context, the approach can be used for tracking opinion-in-the-moment during elections or a public campaign and obtaining information on the public sentiment and their engagement levels. In addition, because it is stable and scalable, it could serve as an excellent candidate for implementing in a large-scale or real-time text analysis system. The WOOSD-CNN model, despite its high accuracy due to multi-feature fusion and attention mechanisms, requires significant computational complexity. Techniques like model pruning and quantization can reduce complexity, making it suitable for real-time processing.

# CONCLUSION

The classification of public opinion sentiment in social media has always been a pivotal area of sentiment analysis, where the quality of emotional word vectors directly impacts the model's categorization performance. In response, firstly, we propose a WOOSD-CNN based on coarse-grained sentiment analysis. The WOOSD-CNN model and the multi-feature fusion strategy used along with an advanced information interaction channel make this study unique as they solve two significant problems in sentiment analysis, including the label drift issue and the problem of boosting the accuracy of multiple classifications of sentiments at the same time. We utilize Word2Vec and Fasttext to capture semantic and word order features. We introduce emoji frequency to measure emoji intensity, building a sentiment similarity matrix to derive emotional features. Our experiments on two datasets demonstrate that the proposed model outperforms similar models in terms of accuracy, precision, and recall, indicating its superior categorization performance. Then, from a fine-grained perspective, we delve into aspect-based sentiment analysis. Unlike previous works that often classify sentiment for specified aspects, which is unrealistic, our approach leverages textual interaction information to mitigate label drift between aspect term subwords. We introduce a model based on an information interaction channel. Our experiments on social media text dataset show that the proposed method enhances text representation, reduces LDP errors to a certain extent, and achieves good performance. In healthcare, the model can be used to assess patient feedback, enabling healthcare providers to improve service quality and patient care based on sentiment trends. The multi-feature fusion approach makes the model adaptable to various domains that require nuanced sentiment analysis across diverse textual data sources.

## Limitations/validity

The proposed mechanisms for addressing label drift and leveraging sentiment intensity transfer weights may struggle with complex or culturally diverse texts, limiting generalizability across languages. Additionally, the multi-feature fusion approach increases the risk of overfitting, particularly in small or imbalanced datasets. Scalability is another concern, as the unified learning framework might face challenges in large-scale or real-time applications due to higher computational costs. The WOOSD-CNN model is computationally expensive, but because of the modularity of its design, this model could be used for real-time applications. This means that pruning techniques, lightweight embeddings, parallel processing, and the utilization of GPUs can optimize it. Subsequent research is to shed light on edge computing settings. Furthermore, the model's dependence on supervised training data means that its performance is contingent on the availability and quality of labeled datasets, which could be a constraint in certain domains.

This study finds that the chosen datasets are valid for assessing the model under the condition of Chinese social media data, and the generalization of the discussed approach to non-Chinese or multilingual data is beyond this investigation. The semantic, word order, and lexicon-based features provide a highly conducive environment that can be extended to include other languages. However, issues like difficulty identifying and procuring high-quality sentiment lexicons and disparity in syntagmatic variations may impede the

students' performance. Future work could be oriented toward expanding the proposed model using multilingual pre-trained embeddings, such as multilingual BERT and language-specific lexicons, to make the proposed model more universal. Apt solutions to these challenges would make the proposed WOOSD-CNN model particularly useful in tackling sentiment analysis across global social media. The multi-feature fusion approach in WOOSD-CNN increases overfitting risks, especially for small or imbalanced datasets. Regularization techniques such as dropout, early stopping, and cross-validation were used to mitigate this risk, with plans to explore data augmentation for enhanced generalizability.

Ethical considerations are crucial when using sentiment analysis models, as bias can lead to skewed predictions and misuse of sentiment analysis. Continuous monitoring and evaluation are essential, with future research focusing on bias reduction strategies by implementing fairness algorithms. Future research could adapt the WOOSD-CNN model for multilingual sentiment analysis using pre-trained embeddings and explore domain-adaptive training to enhance performance in specialized fields like healthcare and finance, increasing its real-world versatility.

### Funding
The study was supported by the key commissioned project for the research on higher education teaching reform in Heilongjiang Province is "Innovation and Practice in Cultivating Cultural and Tourism Composite Talents in Local Universities under the Background of New Liberal Arts" (No. SJGZ20220135). The funders had no role in study design, data collection and analysis, decision to publish, or preparation of the manuscript.

### Grant Disclosures
The following grant information was disclosed by the authors:
Heilongjiang Province is "Innovation and Practice in Cultivating Cultural and Tourism Composite Talents in Local Universities under the Background of New Liberal Arts": SJGZ20220135.

### Competing Interests
The authors declare that they have no competing interests.

### Author Contributions
- Yuan Yao conceived and designed the experiments, performed the experiments, analyzed the data, performed the computation work, prepared figures and/or tables, authored or reviewed drafts of the article, and approved the final draft.
- Xi Chen conceived and designed the experiments, performed the experiments, analyzed the data, performed the computation work, prepared figures and/or tables, authored or reviewed drafts of the article, and approved the final draft.
- Peng Zhang conceived and designed the experiments, performed the experiments, analyzed the data, performed the computation work, prepared figures and/or tables, and approved the final draft.

## Data Availability

The code and raw measurements are available in the Supplemental Files.

The data was obtained from Kaggle: https://www.kaggle.com/datasets/new-york-city/nyc-social-media-usage?resource=download.

## Supplemental Information

Supplemental information for this article can be found online at http://dx.doi.org/10.7717/peerj-cs.2643#supplemental-information.

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
