# Peer review of "Social media network public opinion emotion classification method based on multi-feature fusion and multi-scale hybrid neural network"

_PeerJ Computer Science, doi:10.7717/peerj-cs.2643_

## Round 0.1 · original submission · Major Revisions

Dear Colleagues

The experts have now commented on your article and you will see that they have a lot of concerns to be addressed before we re-consider it, please carefully revise the paper in light of these comments and also do the following

Consider defining technical terms more clearly, such as "TPC task," "label drift phenomenon (LDP)," and "WOOSD-CNN." Acronyms and complex terms should be introduced with brief explanations to improve readability for a broader audience

please provide more justification for the choice of techniques, such as semantic fusion, word order features, and the specific use of WOOSD-CNN
please clearly highlight the novelty of the proposed approach.

Discuss the practical applications of this research in real-world scenarios
Specify the evaluation metrics used to determine accuracy and effectiveness in sentiment polarity categorization
Please improve the language quality of the manuscript

·

Basic reporting

Detailed Comments: The paper provides an interesting approach to sentiment analysis, incorporating multi-feature fusion and hybrid neural networks for public opinion classification in social media. However, some areas need further elaboration, and specific improvements are recommended to enhance clarity and relevance. A major revision is suggested to address the points below:
Comments:
The paper would benefit from a clearer articulation of the research gap in sentiment analysis on social media. More specific points about the limitations of existing methods should justify the proposed approach.

Although model performance metrics are provided, further comparisons with other state-of-the-art models are necessary to establish the novelty and effectiveness of WOOSD-CNN.

Experimental design

More explanation on why specific features (semantic, word order, lexicon-based) were chosen and how their fusion contributes uniquely to model accuracy would strengthen the proposed method's justification.

The selected datasets are appropriate for testing the model, yet it would be valuable to discuss the model’s applicability to non-Chinese or multilingual data, given the universal goal of social media sentiment analysis.

The paper should provide insights into scalability for real-time applications, particularly the computational requirements of WOOSD-CNN and how it might perform on resource-limited devices.

A discussion on computational complexity and its trade-offs in accuracy would be useful, especially regarding real-time processing needs in practical applications.

Validity of the findings

Social media data is often noisy; a more detailed exploration of the noise handling strategies incorporated in data preprocessing would be beneficial.

Exploring the model’s performance on cross-domain or cross-topic sentiment analysis tasks could significantly enhance the scope of this study.

There should be an acknowledgment of potential overfitting risks associated with the model's multi-feature fusion approach, especially for small or imbalanced datasets.

Additional comments

NA

Reviewer 2 ·

Basic reporting

The paper presents an emotion classification approach using a multi-feature fusion and hybrid neural network model, designed to enhance sentiment analysis performance on social media texts. It focuses on capturing semantic and word order features and explores lexicon-based sentiment vectors, proposing a fusion model, WOOSD-CNN, for more accurate classification. Additionally, an aspect-level sentiment analysis model utilizing information interaction channels aims to tackle label drift and improve aspect-term extraction accuracy. The authors demonstrate improved performance on two Chinese microblog datasets using various model comparison metrics. Followings are some the suggestions for the improvement of the work:
1) The paper presents a label drift solution, but additional evaluation on its effectiveness with varying label distributions would provide robustness.
2) A comparison of WOOSD-CNN’s computational efficiency against simpler models such as CNN and RNN would clarify the benefits of its increased complexity.
3) The paper would benefit from detailing the parameter tuning process, especially how hyperparameters were optimized for WOOSD-CNN.
4) A broader discussion on applications outside social media (e.g., customer reviews, feedback in healthcare) could highlight the model’s versatility.
5) Practical challenges in integrating WOOSD-CNN with existing platforms, especially those with privacy concerns, should be addressed.
6) Discussing ethical implications, such as bias in sentiment analysis models and potential for misuse, would be relevant.
7) A brief mention of the specific hardware used in experimentation would help gauge model requirements for potential adopters.
8) Metrics beyond accuracy, such as latency and memory usage, should be included, as they are crucial for real-time applications.
9) The paper needs substantial revisions for language and grammar. Therefore, thoroughly check the manuscript and make corrections.
10) Expanding the conclusion with implications of the study and potential real-world impact of WOOSD-CNN would provide a stronger ending.
11) Including more recent literature on hybrid neural networks in sentiment analysis could enhance the study's contextual relevance.
12) Suggesting specific directions for future research, such as exploring multilingual models or domain-adaptive training, could provide useful context for readers.

Experimental design

All points mentioned part of basic reporting.

Validity of the findings

All points mentioned part of basic reporting.

Additional comments

All points mentioned part of basic reporting.

---

## Round 0.2 · accepted · Accept

Based on the experts' input on the revised manuscript, I am pleased to inform you that your manuscript is being recommended for publication. Congratulations!!!

·

Basic reporting

All the concerns have been addressed. The paper can be accepted in its current state.

Experimental design

NA

Validity of the findings

NA

Additional comments

NA

Reviewer 2 ·

Basic reporting

The authors have addressed my comments and there is no further comment from my side.

Experimental design

Changes done by authors are satisfactory, no further comments.

Validity of the findings

No further comments.

Additional comments

No further comments.
Before submitting the final version for the publication, please have a thorough review of the text to remove any grammatical mistakes and to improve flow of the information.